# Challenges to the Implementation of BIM for the Risk Management of Oil and Gas Construction Projects: Structural Equation Modeling Approach

**Ahsan Waqar** [1,*] , **Idris Othman** [1] **and Roberto Alonso González-Lezcano** [2]

1   Department of Civil & Environmental Engineering, University Technology PETRONAS,
    Seri Iskandar 32610, Malaysia
2   Department of Architecture and Design, Escuela Politécnica Superior, Montepríncipe Campus,
    Universidad San Pablo-CEU, CEU Universities, 28668 Madrid, Spain
*   Correspondence: ahsan_21002791@utp.edu.my

**Abstract:** Building Information Modeling (BIM) has become increasingly popular in the construction industry as a way to enhance risk management. However, little attention has been paid to the challenges of using BIM for safety management in Malaysia's oil and gas construction sector, which is particularly hazardous and requires effective safety management to complete projects successfully. This study aims to identify the obstacles to using BIM for safety management in Malaysia's oil and gas construction sector and to understand the root causes of resistance to its adoption. Exploratory factor analysis and structural equation modeling were conducted on survey data collected from industry professionals. The study found that knowledge obstacles, creative hurdles, technical barriers, supervisory barriers, and functional barriers are the most significant challenges hindering the widespread adoption of BIM for safety management. These challenges were confirmed to significantly affect BIM adoption for safety management. The study's findings have important implications for policymakers, industry practitioners, and academics seeking to improve safety management in Malaysia's oil and gas construction sector through the use of BIM. Future research could explore additional variables that may impact BIM adoption for safety management in this sector.

**Keywords:** Building Information Modeling (BIM); risk management; oil and gas construction; Malaysia; structural equation modeling

## 1. Introduction

The oil and gas industry presents a unique set of challenges, encompassing not only technical complexity and financial risk but also occupational hazards that pose a constant threat to workers' safety. Oil and gas projects need the cooperation of several parties, including engineers, contractors, subcontractors, and regulatory bodies [1,2]. These projects are also very capital-intensive and call for substantial commitments of time and assets. Given their complexity, several risks must be handled to guarantee the effective completion of these projects.

Oil and gas construction projects are susceptible to hazards and mishaps that may have serious safety and financial repercussions [3,4]. The United States Occupational Safety and Health Administration (OSHA) reports that those working on oil and gas construction projects have a higher death rate than the average across all industries. In 2019, the mortality rate for oil and gas construction projects was 9.2 per 100,000 full-time equivalent employees, compared to the construction industry average of 3.5 per 100,000 [1,5]. Construction projects involving oil and gas necessitate operating at height, which may lead to falls and slides. In 2019, 16% of deaths in the oil and gas business in the United States were caused by falls and slips. Powerful machinery and equipment may hit or capture workers [2,6]. In 2019,

21% of all deaths in the U.S. oil and gas business resulted from being hit by or stuck by equipment [7,8].

BIM has evolved as a potent tool for the design, execution, and operation of large-scale construction projects, such as those in the oil and gas sector [9,10]. BIM digitally depicts a building or structure, revealing its physical and functional properties. It allows all stakeholders to cooperate and exchange information throughout a project's planning, construction, and maintenance phases. Using BIM in the oil and gas sector might aid in mitigating building project hazards. Before construction starts, BIM may detect possible design clashes and conflicts, decreasing the chance of rework and delays [11]. BIM-based risk management refers to the use of Building Information Modeling (BIM) technology to identify, assess, and mitigate potential risks throughout the entire lifecycle of a construction project [9,10]. BIM provides a collaborative platform that enables stakeholders to visualize and analyze data, and make informed decisions that enhance project outcomes and reduce risk [12,13]. BIM-based risk management can involve various approaches such as 4D/5D modeling, clash detection, and virtual design and construction [12]. These techniques can help identify potential issues, mitigate risks, and improve project outcomes [13].

Despite the potential advantages of BIM, its use in the oil and gas sector is still in its infancy. There are challenges to implementing BIM, including requiring specialized software, qualified employees, and stakeholder coordination [14,15]. Nevertheless, as the advantages of BIM become more generally acknowledged, its application in the oil and gas sector is anticipated to rise [16,17].

A potential research gap in the field of BIM implementation for risk management in oil and gas construction projects in Malaysia is the need for studies employing a comprehensive structural equation modeling approach to identify the interrelationships between the various factors that impede BIM adoption [18,19]. Although prior studies have investigated the hurdles and difficulties of BIM adoption in the Malaysian construction industry, only some have studied the particular problems associated with the oil and gas sector and those that still require statistical analysis. In addition, the function of BIM in risk management within the Malaysian oil and gas sector has received less attention [20,21]. This paper attempts to solve this research gap using a structural equation modeling technique to examine the barriers to BIM adoption for risk management in Malaysian oil and gas construction projects.

The paper discusses the challenges associated with deploying BIM risk management in oil and gas construction projects in Malaysia. The oil and gas sector is capital-intensive and calls for substantial commitments of time and assets, and several parties need to cooperate to ensure effective project completion. Construction projects in this sector are susceptible to hazards and mishaps that may have serious safety and financial repercussions. BIM, as a tool for the design, execution, and operation of large-scale construction projects, may aid in mitigating building project hazards, detecting possible design clashes and conflicts, and increasing overall efficiency and effectiveness. However, there are challenges to implementing BIM, including requirements involving specialized software, qualified employees, and stakeholder coordination. The study aims to identify the interrelationships between the various factors that impede BIM adoption in Malaysia's oil and gas sector and present ideas for overcoming these challenges using a comprehensive structural equation modeling approach. The paper contributes to the literature on BIM implementation for risk management in the Malaysian oil and gas sector.

It should be noted that whereas the routine procedures of risk management in the oil and gas industry have been established, there is a lack of research on the potential impact of BIM technology on enhancing risk management procedures. Therefore, this study seeks to fill this gap by applying a structural equation modeling technique to comprehensively examine the interrelationships between the constraints that impede BIM adoption for risk management in the Malaysian oil and gas sector. The unique method proposed in this study aims to identify and overcome these challenges to enhance risk management in the oil and

gas construction industry, which may have implications for other settings and industries as well.

Overall, this study contributes to the literature on risk management in the oil and gas construction industry by applying a comprehensive structural equation modeling technique to investigate the challenges associated with BIM adoption for risk management in Malaysia. The study identifies the interrelationships between various constraints that impede BIM adoption and examines the potential impact of BIM technology in enhancing risk management procedures. The findings of this study provide a unique method for identifying and overcoming the challenges of using BIM for risk management in the Malaysian oil and gas construction business and may have relevance for other industries and settings.

## 2. Current Risk and Safety Management Concerns

In 2019, the Malaysian Department of Occupational Safety and Health (DOSH) recorded 120 incidents in the oil and gas industry. These incidents caused ten deaths, eleven permanent impairments, and seventy-four temporary disabilities. The DOSH also noted that falls were the leading cause of accidents in the Malaysian oil and gas sector, followed by struck-by and caught-between incidents [22]. The Department of Occupational Safety and Health has stressed the need for industry enterprises to strengthen their safety management systems and practices to avoid accidents and enhance safety performance [23,24]. According to research conducted by Al-Mutairi & Younes, and AlMarar, on safety practices in the Malaysian oil and gas industry, insufficient safety training, poor safety culture, and inadequate safety management systems are key challenges to improving safety performance in the sector. The report advised creating a complete safety management system with effective risk management, safety leadership, and routine worker safety training [25,26].

According to research by the International Organization of Oil and Gas Producers (IOGP), 30 fatal incidents occurred in the upstream oil and gas sector globally in 2019. In addition, 81 nonfatal accidents resulted in at least one day off work or limited tasks [27]. The IOGP survey also noted that slips, trips, and falls were the leading causes of accidents in the oil and gas sector, followed by struck-by and caught-between incidents. The research stressed the need for a strong safety culture, an effective risk assessment, and robust safety management systems to avoid accidents and enhance the industry's safety performance [12,13].

In 2019, there were 98 fatal incidents in the oil and gas extraction business in the United States, with a mortality rate of 9.2 per 100,000 full-time equivalent employees, according to a study by the Bureau of Labor Statistics (BLS) [28,29]. According to the BLS data, the primary causes of fatal workplace accidents include transportation mishaps, falls, and contact with items and equipment. According to these figures, the oil and gas sector is a high-risk business requiring excellent safety management systems and procedures to avoid accidents and safeguard employees.

The literature indicates that the complexity and heterogeneity of the oil and gas construction industry present a significant challenge for implementing BIM for risk management [30,31]. The use of BIM requires standardization of processes and collaboration across multiple stakeholders, which can be difficult to achieve in this industry [31]. Additionally, the unique safety and regulatory requirements of oil and gas construction projects may require customization of BIM applications, adding to the complexity of implementation.

Moreover, measuring and defining risk factors in the context of oil and gas construction projects can also be challenging. Several studies have identified the need to develop a comprehensive framework for risk management that integrates BIM data with other project data sources to provide a holistic view of project risk [32].

Finally, the successful implementation of BIM for risk management in oil and gas construction projects requires a significant investment in technology, training, and expertise [33]. This highlights the importance of management support and commitment to BIM implementation, and the need for a skilled workforce that can leverage the benefits of BIM.

In conclusion, the literature highlights the need for standardization, collaboration, comprehensive risk management frameworks, and management support to successfully implement BIM for risk management in oil and gas construction projects. The SEM approach can provide insights into the interrelationships between various factors affecting BIM implementation for risk management in this industry.

## 3. BIM for Risk Management

BIM is a digital technology that may improve oil and gas construction safety management and risk reduction. BIM offers a collaborative design, construction, and operation platform, enabling real-time data and information sharing among project stakeholders. Providing a 3D model of the construction site is one way that BIM may aid in the risk management of oil and gas development projects [34,35]. The 3D model may simulate and assess possible safety dangers, such as equipment collisions, fall risks, and collision risks. This enables project managers and safety staff to identify possible dangers and create effective safety procedures to reduce them. Numerous prior studies have investigated the potential advantages of BIM for enhancing safety management and risk reduction in the construction industry, particularly the oil and gas industry [36,37]. Elwany & Elsharkawy [38] and Mohd Hanafiah et al. [32], for example, investigated the potential of BIM for addressing health and safety concerns in building projects. The research discovered that BIM might assist in detecting, evaluating, and reducing safety concerns by offering a collaborative platform for stakeholders to exchange data and information in real time [33,39].

Annamalah et al. examined the potential for BIM to improve the safety performance of construction projects. The research determined that BIM may be used to simulate and assess possible safety dangers, enabling project managers and safety specialists to detect potential risks and devise effective safety measures to minimize such risks. In addition, Annamalai et al. and Jagoda & Wojcik investigated the potential for BIM to improve the safety of oil and gas construction projects in the United Arab Emirates [38,40]. The researchers discovered that BIM might be used to detect and reduce safety hazards throughout the design and construction stages of oil and gas projects, enhancing safety performance and decreasing the number of accidents and injuries. These studies illustrate the capability of BIM to improve safety management and risk reduction in the construction industry, particularly the oil and gas industry [30,31]. BIM can assist in discovering, evaluating, and mitigating safety hazards by offering a collaborative platform for stakeholders to exchange data and information in real time, enhancing safety performance and lowering the number of accidents and injuries. BIM also facilitates the adoption of safety management systems by giving real-time data on the project's safety performance [32,41]. By incorporating safety performance data into the BIM platform, project stakeholders may track safety metrics like the number of safety occurrences, safety violations, and safety training compliance. This enables project managers to recognize safety patterns and implement corrective measures to enhance safety performance.

The complexity and heterogeneity of the oil and gas construction sector are two of the main problems mentioned in the literature. Contractors, subcontractors, suppliers, and regulators are just a few of the many players in the sector, many of whom sometimes work in isolation. In order to adopt BIM, various stakeholders must collaborate and procedures must be standardized, which may be challenging in this sector. According to research by Yan et al. and Derakhshanalavijeh & Teixeira, the sector should create standardized BIM implementation processes and foster a collaborative culture that promotes information exchange and collaboration among stakeholders [17,18].

Another issue is the absence of a thorough framework for risk management that combines BIM data with other project data sources to provide a full picture of project risk. Such a framework has been recommended by several studies to facilitate efficient risk detection, analysis, and reaction planning. To improve safety and risk management in oil and gas construction projects, Said et al. developed a system that combines BIM data with hazard identification and risk assessment approaches [14].

It might be difficult to measure and define risk variables in the context of oil and gas building projects. According to the research, BIM data alone may not provide a comprehensive picture of project risk, since BIM models might not account for certain risk elements, such as human factors, organizational culture, and outside events [11,15]. Therefore, a thorough risk management system should include additional data sources and integrate expert and stakeholder evaluations of the risks.

Last but not least, a significant investment in technology, training, and knowledge is needed to apply BIM for risk management in oil and gas construction projects. Numerous studies have emphasized the need for building a competent workforce that can take advantage of BIM's advantages, as well as the necessity of management support and dedication to BIM adoption [8,42,43]. The establishment of training programs that concentrate on BIM implementation in oil and gas construction projects as well as the creation of a BIM maturity model that allows companies to gauge their BIM implementation progress are both essential, according to research by Isnadi et al. (2022) [1].

Moreover, BIM may enhance communication and cooperation between project stakeholders, such as the design team, contractors, and safety officials [44,45]. BIM can offer a single platform for project stakeholders to communicate safety-related information and data, enabling them to identify safety hazards and design effective safety solutions in collaboration. BIM may be a useful risk management technique for oil and gas building projects. BIM may enhance safety performance, decrease safety hazards, and allow effective safety management systems by offering a collaborative design, construction, and operation platform.

## 4. Methodology

This research aimed to analyze and identify the barriers impeding BIM application for safety management in the Malaysian oil and gas construction sector. Therefore, an exploratory research design was adopted to perform the critical literature evaluation, and many stages of data collection and organization were used. A critical review reveals that Ajmal, Bin Isha, et al. [46], and Leth et al. [47], have conducted a thorough study and critical analysis of previous research; it goes beyond presenting well-known publications to incorporate fresh ideas and views [48,49]. This study's data was collected by evaluating several sources, including published articles, research papers, government documents, and green building regulations, to achieve a comprehensive picture. Then, previous research's data analysis, refinement, and classification were summarized [50]. Literature review is important for identifying challenges in the implementation of BIM for the risk management of oil and gas construction projects because it allows researchers to gain a comprehensive understanding of the existing knowledge and gaps in the field. This helps to identify key challenges and research questions and inform the development of research methodology and data analysis. As part of the data review process, the obtained data must be trimmed down via selection, simplification, and data abstraction. From the literature review on challenges to BIM-based safety and risk program implementation, a collection of 24 barriers was derived and judged suitable for constructing the PLS-SEM model. [51,52]. Several studies on the PLS-SEM approach have recently appeared in prominent SSCI journals. The most current version of the software SMART-PLS 4 was used to analyze the acquired data in order to use SEM to estimate the importance of the challenges to BIM-based safety program implementation [53,54]. PLS-SEM was first lauded for its superior prediction skills over covariance-based structural equation modeling (CB-SEM), although there are only minor differences between the two techniques. This study's statistical analysis includes measurement and a structural model assessment method. The study design (Figure 1) indicates all stages involved in the analysis.

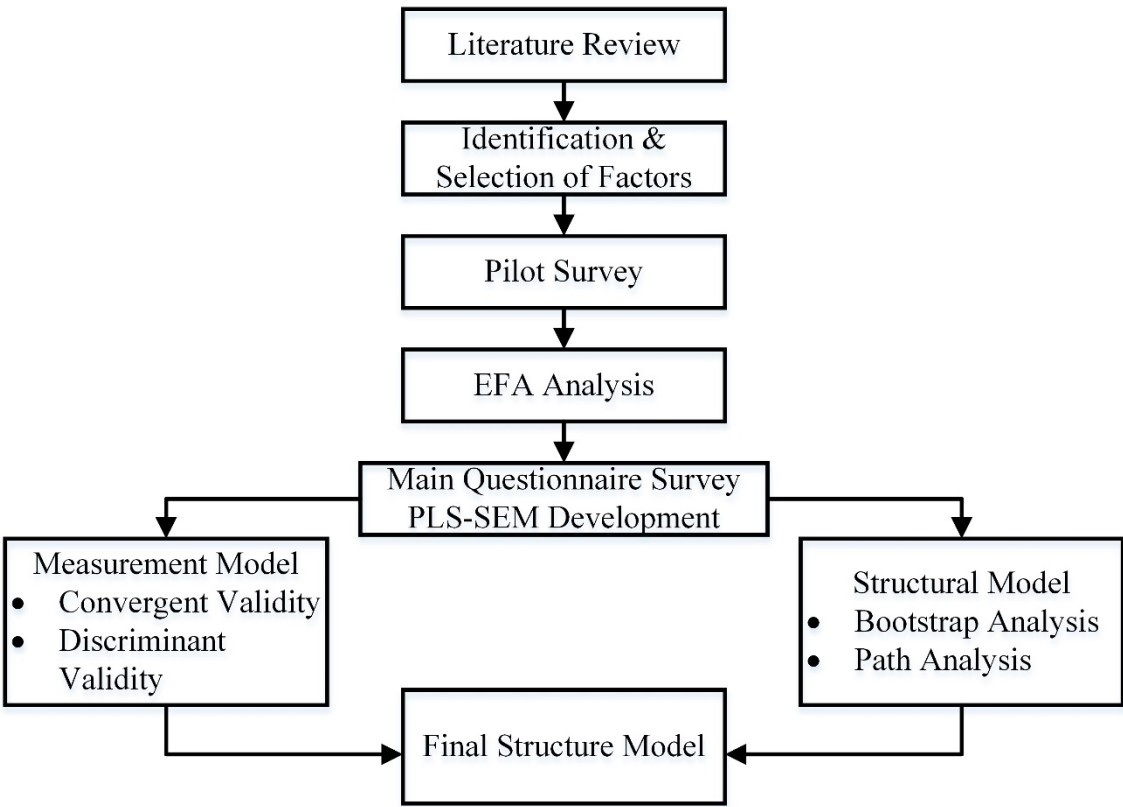

**Figure 1.** Research methodology flowchart.

*4.1. Data Collection*

Contact was made with a wide range of possible Malaysian oil and gas industry players in the business to study the implementation issues for BIM initiatives for risk management. The survey was split into three sections: the demographic features of the claimant, the process BIM uses, and open-ended questions (to include any challenges the users felt were necessary to be mentioned) [55,56]. Clients, consultants, and contractors were the three primary target populations contacted. The professions or vocations of designers, electricians, cost engineers, construction managers, and manufacturers might be further subdivided. Respondents rated implementation challenges for BIM initiatives using a 5-point Likert scale, with five being very high, four representing high, three representing medium, 2 representing moderate, and one representing nil or very little. In prior studies, this scale was used. Since BIM-based safety and risk management is still relatively new in Malaysia, the sampling process for the relevant subpopulation was considered [57,58].

Similarly, a methodological purpose analysis was used to determine the sample size for this investigation. According to Mohd Hanafiah et al. [39], the sample size must exceed 100 to employ SEM. Due to using the SEM technique, 155 out of 210 were contacted for this research, yielding a response rate of around 73%. This rate of return was regarded as adequate for this kind of experiment, based on prior studies.

*4.2. Exploratory Factor Analysis*

EFA (Exploratory Factor Analysis) is a statistical method for determining the underlying structure of a data collection. In EFA, a researcher investigates the data to uncover the underlying factors that explain the variance in the observable variables. EFA yields a collection of factors, each with a factor loading value that reflects the strength of each variable's association with each factor [59,60]. The range of the factor loading value is from -1 to 1. A number closer to 1 suggests a strong relationship between the variable and the factor, whereas a value closer to 0 indicates a poor relationship [46,47]. A negative number implies an inverse relationship between the variable and the factor and also offers

information on the eigenvalue, percentage of variance explained, and commonalities in addition to factor loading values [61,62]. The eigenvalue measures the amount of variation in the observable variables that is explained by each component [42,43]. Factors with eigenvalues of more than 1 are considered important. This is the percentage of observed variation explained by each component. Commonalities assess the percentage of variation in each observable variable that is explained by the combination of all the causes.

In conclusion, EFA offers the following values:

- Factor loading values (ranging from −1 to 1)
- Eigenvalues
- Explanation of the variance proportion
- Commonalities

By analyzing these values, researchers may gain insights into the underlying structure of the observed variables and find the elements that underlie the data's volatility. The preceding described data classification into significant groups or concepts [63]. This classification has been accomplished by ensuring that each piece of information is assigned to the appropriate subgroups (subconstructs) of key constructs.

### 4.3. Measurement Model

The measuring model exposes the present relationship between items and their hidden structure. The subsequent subsections comprehensively examine the discriminant and convergent validity of the measurement model.

Convergence Validation

Convergent validity is the degree to which two distinct measurements of the same concept are associated. It evaluates whether two distinct methods for determining the same variable provide comparable findings. This is essential in research, since it helps determine the reliability of a measurement instrument and assures that it is detecting what it is designed to measure [64,65]. Convergent validity is the degree of agreement between two or even more measurements (barriers) of the same concept (category) [48]. It adds considerably to the validity of the concept. Cronbach's alpha (ca), composite reliability scores, and average variance extracted (AVE) might be used to measure the convergent validity of the generated constructs in the case of a model. AlNoaimi & Mazzuchi, and Kadam, recommended a composite reliability value of 0.7, since 0.7 was regarded as the threshold for "moderate" composite dependability [66,67]. For all study forms, values more than 0.60 were deemed adequate.

### 4.4. Discriminant Validation

Discriminant validity indicates that, since the phenomenon under research is empirically unique, no measures can accurately identify it. Samimi and Van Thuyet et al. claimed that measurements should not be too similar to ensure discriminating validity. Discriminant validity refers to the degree to which a measure is unique from other measures designed to examine other constructs. It assures that the measurement instrument is not measuring the same construct under various names and is essential for maintaining the correctness of study results [68,69]. Heterotrait-Monotrait (HTMT) analysis, cross-loading, and the Fornell–Larcker criteria are some approaches used to determine discriminant validity. Using the HTMT ratio is one way to determine the discriminant validity of a test. The HTMT ratio compares the correlations between two constructs to the correlations between each construct's elements. If the HTMT ratio exceeds a certain threshold (often 0.85), it indicates a lack of discriminant validity, suggesting that the constructs may be too similar or overlap [70,71]. Cross-loading is another technique examining whether an item loads successfully onto a different construct than its parent build. If an item loads well onto another construct, it shows that it is not particular to its parent construct and may be measuring a distinct construct [72,73]. This raises concerns about the item's discriminant validity. The Fornell–Larcker criterion is an additional frequently used approach

for evaluating discriminant validity [74,75]. This criterion demands that the square root of each construct's average variance extracted (AVE) be greater than its association with another construct. In addition, each item should load most heavily on its corresponding build instead of loading comparably on many structures.

*4.5. Analysis of Structural Models*

This work used SEM to estimate the significance of implementation restrictions for BIM-based risk management initiatives. The model parameters between the measured components must be found to do this. As shown by Equation (1), the structure of the formulae for £, μ, and €1 that was recognized as the inherent link can be expressed mathematically:

$$\mu = £ + €1 \tag{1}$$

where (β) is the route coefficient connecting BIM implementation hurdle constructions and (€) is expected to represent the residual variation at this structural level. It represents the standardized regression weight, corresponding to the multiple regression model's weight [76,77]. A clear indicator must be statistically significant and consistent with the model's expectations [78,79]. Determining the relevance of the route coefficient is the current topic. As with CFA, the average errors of the route coefficients were computed using an implementation technique in the SmartPLS 4 application. Quintino et al. determined the t-statistics used in the propositional analysis, and 5000 subsamples were used per their suggestion [2]. To illustrate the inherent links between the ideas and formulae, four structural equations characterizing the PLS Model's BIM implementation hurdles were constructed (1).

*4.6. Model Validation Survey*

A brief survey questionnaire was utilized to verify the generated structural model. The validation survey included the primary stakeholders of this research, such as safety managers, contractors, and consultants. The objective of validation was to establish the practical applicability of the generated structural model so that suitable actions could be performed to control the variables and help address obstacles and their influence on the application of BIM for safety management in the oil and gas construction sector [66,67]. The authors concur that the validation procedure is essential to the success of this research. Twenty experts were requested to participate in the validation survey, and five critical questions were developed to determine the model's validity.

Q1: Are the factors proposed in the model applicable to obstacles associated with applying BIM for the risk management of oil and gas construction projects in Malaysia?

Q2: Is the model reasonable for identifying the critical barriers affecting the application of BIM for the risk management of oil and gas construction projects in Malaysia?

Q3: Are the factors presented in the structural model reasonable for obstacles associated with the application of BIM for the risk management of oil and gas construction projects in Malaysia?

Q4: Do you find the study results reasonable?

Q5: Can the structural model presented in the study be generalized?

## 5. Identification of Challenges

Several challenges must be considered when planning to use Building Information Modeling (BIM) in oil and gas building projects. As in Table 1, The lack of standardization in the safety data generated by BIM is a key obstacle (B1). This might make deciphering safety data and making well-informed management choices challenging. Adopting BIM promptly and efficiently might be difficult because of resistance to change (B2). Inadequate BIM implementation for risk management is a further difficulty caused by the absence of standards (B3). Integration might be challenging as BIM software only minimally interfaces with other systems (B4). Further challenges to implementation include (B5) a lack of readily

available hardware that is compatible with the necessary BIM software, (B6) the demand for continual monitoring, and (B7) a dearth of readily available BIM professionals to provide guidance and help (B7). The challenges described in Table 1 were obtained through a literature review.

**Table 1.** Identified challenges of BIM in oil and gas construction projects.

| Factors | Description | References |
| --- | --- | --- |
| B1 | Absence of data uniformity for safety data produced by BIM. | [20,24] |
| B2 | Opposition to change. | [80,81] |
| B3 | Low use of BIM for safety management due to the need for more standardization. | [2,28] |
| B4 | BIM software only partially interacts with other software systems. | [39,82] |
| B5 | Limited access to hardware meeting BIM software requirements. | [77,83] |
| B6 | The need for constant surveillance. | [7,14] |
| B7 | Restricted accessibility to BIM specialists for advice and assistance. | [27,31] |
| B8 | Implementing BIM is expensive. | [1,63] |
| B9 | Language differences. | [2,28] |
| B10 | Restricted availability of BIM training programs. | [15,23] |
| B11 | Knowledge of BIM applications for safety management. | [27,30] |
| B12 | Managing BIM processes across various stakeholders may be challenging. | [31,32] |
| B13 | Minimal usage of BIM for post-construction safety management. | [3,33] |
| B14 | Inadequacies in the capture of data onsite. | [51,75] |
| B15 | Integration with current systems. | [18,84] |
| B16 | Safety and false alarms. | [3,33] |
| B17 | Poor coordination amongst stakeholders for BIM implementation. | [80,81] |
| B18 | The need for high-speed cyberspace availability for cloud-based BIM software. | [6,85] |
| B19 | Limited use of BIM for maintaining and running oil and gas facilities. | [2,28] |
| B20 | Risk and supervision complications associated with oil and gas industries. | [16,82] |
| B21 | The need for technical safety integration. | [1,2] |
| B22 | Socioeconomic concerns. | [20,24] |
| B23 | Inadequate technology execution scope. | [16,21] |
| B24 | Data discrepancy. | [28,39] |

The cost of implementing BIM (B8) might be prohibitive for businesses that are already strapped for cash. B10: A lack of readily accessible BIM training programs may reduce the pool of qualified specialists available to work with BIM on oil and gas construction projects, which in turn can exacerbate B9: Linguistic challenges. Inadequacies in onsite data capturing (B13) and a lack of familiarity with BIM software for safety management (B11) are two other problems (B14). Safety concerns, false alarms (B16), and a lack of stakeholder collaboration might slow the BIM adoption process (B17). The limited usage of BIM in the maintenance and operation of oil and gas facilities (B19) and the risk and supervisory problems connected with the oil and gas sectors may also be barriers to adoption, as might the lack of high-speed internet access for cloud-based BIM software (B18) (B20). Inadequate technology execution scope (B21), socioeconomic problems (B22), and a lack of safety technology integration (B21) might all be problematic (B23). Ultimately, it might be

challenging to verify that data is correct and dependable while dealing with BIM due to data discrepancy (B24). These challenges show how much thought must be given to using BIM in oil and gas building projects before the advantages of BIM can be realized.

## 6. Results

### 6.1. Exploratory Factor Analysis (EFA)

Five components with eigenvalues larger than one were found after the investigation [84,86]. Challenges with BIM implementation, such as the requirement for ongoing monitoring, limited access to BIM professionals, and the high cost of implementation, are grouped under Construct 1′s B23, B11, B4, B7, B1, and B6. Complications with risk and supervision in the oil and gas business and a need for more technology integration for safety make up Construct 2. Thirdly, the absence of standards, high hardware costs, and a dearth of BIM training programs all contribute to the technology's low adoption rates in Construct 3 (B3, B8, B19, and B5, respectively). Construct 4 consists of B14, B15, and B13, all of which have to do with insufficient data collection at the job site, a lack of connection of BIM with existing systems, and the infrequent use of BIM for post-construction safety monitoring. The difficulties in managing BIM procedures amongst various stakeholders and issues with safety and false alarms are addressed in factors B12 and B16 of Construct 5, respectively. A high degree of internal consistency is shown by Cronbach's alpha values for Constructs 1, 2, and 3, suggesting that the problems in these areas are interconnected.

Nevertheless, Cronbach's alpha values are somewhat lower for Constructs 4 and 5, suggesting that difficulties in these areas may not be as tightly associated. Since their loading was less than 0.5, our analysis did not include tests for challenges B2, B9, B10, B17, B22, and B24. In conclusion, the EFA findings shed light on the underlying structure of the stated problems of BIM in oil and gas construction projects, and may be utilized to influence the creation of effective ways to overcome these issues. The Table 2 displays the rotated component matrix, eigenvalues, and the proportion of variation described by each factor.

All five constructs evident from the EFA analysis were renamed, such as, Construct 1 as "knowledge barriers," Construct 2 as "technical barriers," Construct 3 as "creativity barriers," Construct 4 as "functioning barriers," and Construct 5 as "supervision barriers." The EFA findings have revealed many knowledge hurdles to using BIM in oil and gas construction projects. Inadequate technological execution scope (B23), lack of understanding of BIM applications for safety management (B11), limited interaction of BIM software with other software systems (B4), and limited access to BIM professionals for guidance and help all figure prominently among these challenges (B7). These constraints may restrict the oil and gas industry's capacity to reap the full advantages of BIM technology, which might impede the effective adoption of BIM. For this reason, it is crucial to raise awareness of BIM applications for safety management and provide sufficient training and support for BIM professionals to break down the existing knowledge barriers [80,87]. Making BIM more compatible with other programs is also important for easy data transfer and integration.

Inadequacies in data capturing onsite (B21), insufficient use of BIM for post-construction safety monitoring (B13), and lack of connection with existing systems (B15) are some of the technological impediments revealed by the EFA findings (B14). Limitations in integrating BIM data with current systems and difficulties in collecting and successfully utilizing data imply that adopting BIM for safety management in the oil and gas construction sector is difficult owing to technological hurdles. To guarantee that BIM data is properly integrated into safety management processes, increased cooperation between technical specialists and safety managers is required, as shown by the absence of safety technical integration [81,82]. There may also be a need for more knowledge or understanding of how BIM may be utilized to enhance safety outcomes once construction is complete, as seen by the limited use of BIM for post-construction safety management. Finally, BIM's efficacy in this setting may need to be improved in regard to gathering and interpreting reliable data to make safety management choices due to shortcomings in onsite data collection.

**Table 2.** Rotated component matrix.

| Factors | Component | | | | | Cronbach Alpha |
|---|---|---|---|---|---|---|
| | 1 | 2 | 3 | 4 | 5 | |
| B23 | 0.816 | | | | | 0.866 |
| B11 | 0.767 | | | | | |
| B4 | 0.713 | | | | | |
| B7 | 0.611 | | | | | |
| **B17** | | | | | | |
| B21 | | 0.761 | | | | 0.781 |
| B13 | | 0.705 | | | | |
| B15 | | 0.681 | | | | |
| B14 | | 0.616 | | | | |
| **B24** | | | | | | |
| B20 | | | 0.839 | | | 0.770 |
| B3 | | | 0.809 | | | |
| B8 | | | 0.711 | | | |
| B19 | | | 0.610 | | | |
| **B2** | | | | | | |
| **B10** | | | | | | |
| B5 | | | | 0.788 | | 0.719 |
| B18 | | | | 0.689 | | |
| B1 | | | | 0.624 | | |
| **B22** | | | | | | |
| B12 | | | | | 0.654 | 0.701 |
| B16 | | | | | 0.619 | |
| B6 | | | | | 0.601 | |
| **B9** | | | | | | |
| Eigenvalue | 3.511 | 3.011 | 2.973 | 2.195 | 2.110 | |
| % Variance | 15.156 | 13.116 | 12.529 | 10.161 | 9.994 | |

Note: Factor B2, B10, B17, B22, B24, B9 excluded from the EFA because of loading less than 0.5.

The high costs of implementing BIM, its limited utility for managing safety, and its limited use in maintaining and operating oil and gas facilities are some recognized impediments to creative thinking. These elements can influence the oil and gas industry's decision to use BIM. According to the EFA findings, these variables significantly negatively impact BIM adoption, since they load heavily on factors 1, 3, and 4. The high expense of adopting BIM (B8) and the absence of standards (B3) may discourage its use by SMEs. The long-term viability of BIM in the oil and gas sector may be jeopardized by its limited adoption in operations and maintenance (B19). In conclusion, BIM adoption in the oil and gas sector requires a risk management strategy due to risks and supervision problems (B20).

Functional challenges are the real-world challenges that prevent BIM from being fully used in oil and gas building projects. One of the biggest technological hurdles is the need for more readily available hardware to run BIM software (B5). Another issue that must be addressed the limited availability of high-speed internet for cloud-based BIM programs (B18). A substantial obstacle must be overcome to guarantee the accuracy of the information collected: more data homogeneity for safety data supplied to BIM (B1). If these problems

are not resolved, BIM cannot do its job on oil and gas construction projects, which might lead to insufficient data integration, poor coordination, and bad decisions [85,88].

Due to the complexity of coordinating and working with many parties participating in the building project, one of the highlighted supervisory hurdles is the difficulty in managing BIM processes across diverse stakeholders (B12). The need for continuous monitoring (B6) may further raise the effort and resources needed to deploy BIM. Due to the oil and gas industry's stringent safety regulations, false alarms and safety issues (B16) may impede efficient BIM deployment. For this reason, it is essential that the BIM system can reliably detect and handle any safety issues without triggering any extra false alarms. Table 3 presents the constructs with all challenges.

**Table 3.** Constructs formulated from EFA along with challenges.

| Constructs | Assigned Code | Challenges |
|---|---|---|
| Knowledge Barriers | B23 | Inadequate technology execution scope. |
| | B11 | Knowledge of BIM applications for safety management. |
| | B4 | BIM software only partially interacts with other software systems. |
| | B7 | Restricted accessibility to BIM specialists for advice and assistance. |
| Technical Barriers | B21 | The need for technical safety integration. |
| | B13 | Minimal usage of BIM for post-construction safety management. |
| | B15 | Integration with current systems. |
| | B14 | Inadequacies in the capture of data on site. |
| Creativity Barriers | B20 | Risk and supervision complications associated with oil and gas industries. |
| | B3 | Low use of BIM for safety management due to the need for more standardization. |
| | B8 | Implementing BIM is expensive. |
| | B19 | Limited use of BIM for maintaining and running oil and gas facilities. |
| Functioning Barriers | B5 | Limited access to hardware meeting BIM software requirements. |
| | B18 | The need for high-speed cyberspace availability for cloud-based BIM software. |
| | B1 | Absence of data uniformity for safety data produced by BIM. |
| Supervision Barriers | s | Managing BIM processes across various stakeholders may be challenging. |
| | B16 | Safety and false alarms. |
| | B6 | The need for constant surveillance. |

The hypotheses developed indicate that the highlighted challenges have a major bearing on the use of BIM for risk management in oil and gas building projects in Malaysia. These hypotheses provide the groundwork for further study and may direct the design of interventions to overcome the challenges indicated. It is crucial to learn what causes these roadblocks and what can be done about them [80,83]. These challenges must be overcome for BIM to be successfully implemented in Malaysia, which would improve the safety and risk management of oil and gas construction projects. To successfully use BIM for

risk management, the hypotheses emphasize tackling numerous challenges concurrently rather than concentrating on a single aspect. The hypotheses developed here provide a hypothesized framework (Figure 2) for SEM analysis in the study of BIM adoption in the context of Malaysian oil and gas building projects. Following are the five hypotheses relevant to five formative constructs.

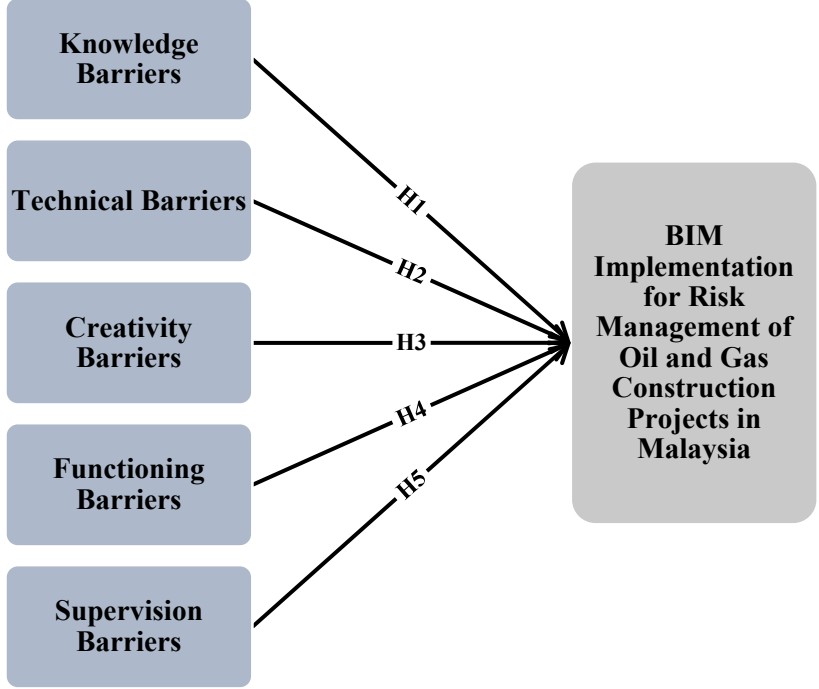

**Figure 2.** Hypothesized framework based on EFA results.

- H1: Challenges in the knowledge barriers construct strongly affect BIM implementation for the risk management of oil and gas construction projects in Malaysia.
- H2: Challenges in the technical barriers construct strongly affect BIM implementation for the risk management of oil and gas construction projects in Malaysia.
- H3: Challenges in the creativity barriers construct strongly affect BIM implementation for the risk management of oil and gas construction projects in Malaysia.
- H4: Challenges in the functioning barriers construct strongly affect BIM implementation for the risk management of oil and gas construction projects in Malaysia.
- H5: Challenges in the supervision barriers construct strongly affect BIM implementation for the risk management of oil and gas construction projects in Malaysia.

*6.2. Demographics*

According to the survey's primary questionnaire, the majority of respondents (61%) had a master's degree, followed by those with a bachelor's degree (19%), and those with a Ph.D. (11%) were in third place. Regarding years of experience, 45% of respondents had 11–15 years, while 20% had 5–10 years. Nine percent of those polled had fewer than five years of professional experience. The majority of responders (54%) were engineers of some kind, followed by those in project management (18%) and then those in the architectural field (10%). Eight percent were safety managers, whereas ten percent were experts in other fields. Civil engineers and project managers, who are frequently involved in the preparation and execution of oil and gas construction projects, appear to have been the primary targets of the survey, as their profiles match those with extensive experience and education in the construction industry. With many survey takers holding advanced degrees, it is safe to assume that those who answered our BIM deployment and risk management

questions are well-versed pros. Figure 3 presents the comprehensive demographic profile of respondents.

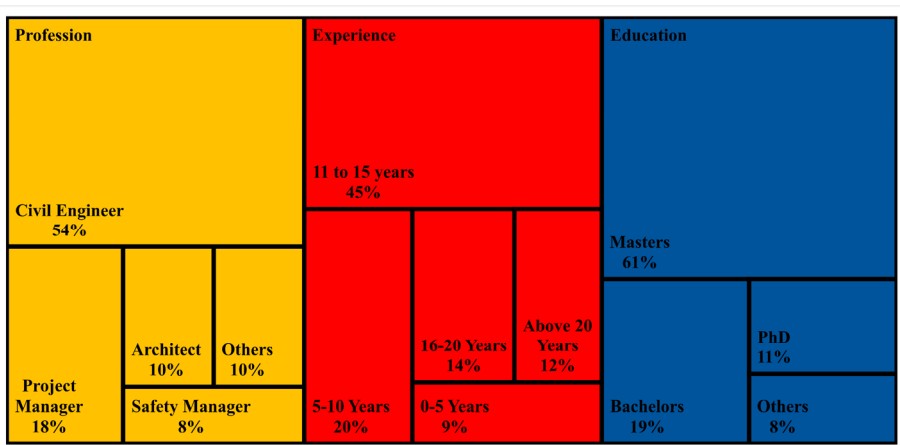

**Figure 3.** Demographic profiles of participants.

### 6.3. Structure Equation Modeling (SEM)

The validity and reliability of the measurement model's five constructs are shown in Table 4. The dependability and consistency of the components inside each construct may be calculated using Cronbach's alpha. Cronbach's alpha values for all constructs are above the acceptable level of 0.7, suggesting that the items reliably assess the same underlying construct. Internal consistency and reliability may also be assessed using composite reliability (rho-a and rho-c), which accounts for intercorrelations between items. All constructs have composite reliability scores greater than the cutoff value of 0.7, indicating strong dependability. Compared to the measurement error, the amount of variation collected by the construct is measured by the average variance extracted (AVE). AVE is considered optimal with a value of 0.5, indicating that the construct captures at least 50% of the variation. All constructs have been validated with an AVE greater than the minimum required. With a Cronbach's alpha of 0.838, composite reliability (rho-a) of 0.847, composite reliability (rho-c) of 0.925, and an average validity estimate (AVE) of 0.86, supervision barriers are the most reliable concept in this measurement paradigm. This suggests that the items used to measure supervision barriers have a high degree of internal consistency and capture a significant percentage of the variation present in the concept [5,63]. Figure 4 indicates the model's overall trend of reliability and validity statistics. The constructed model after PLS algorithm analysis for the measurement model is indicated in Figure 5. Path coefficients of model variables can be seen with positive outcomes on the latent variable.

**Table 4.** Model reliability and validity.

| Constructs | Cronbach's Alpha | Composite Reliability (rho-a) | Composite Reliability (rho-c) | The Average Variance Extracted (AVE) |
|---|---|---|---|---|
| Creativity Barriers | 0.779 | 0.785 | 0.859 | 0.606 |
| Functioning Barriers | 0.703 | 0.778 | 0.866 | 0.765 |
| Knowledge Barriers | 0.71 | 0.719 | 0.837 | 0.632 |
| Supervision Barriers | 0.838 | 0.847 | 0.925 | 0.86 |
| Technical Barriers | 0.82 | 0.831 | 0.893 | 0.735 |

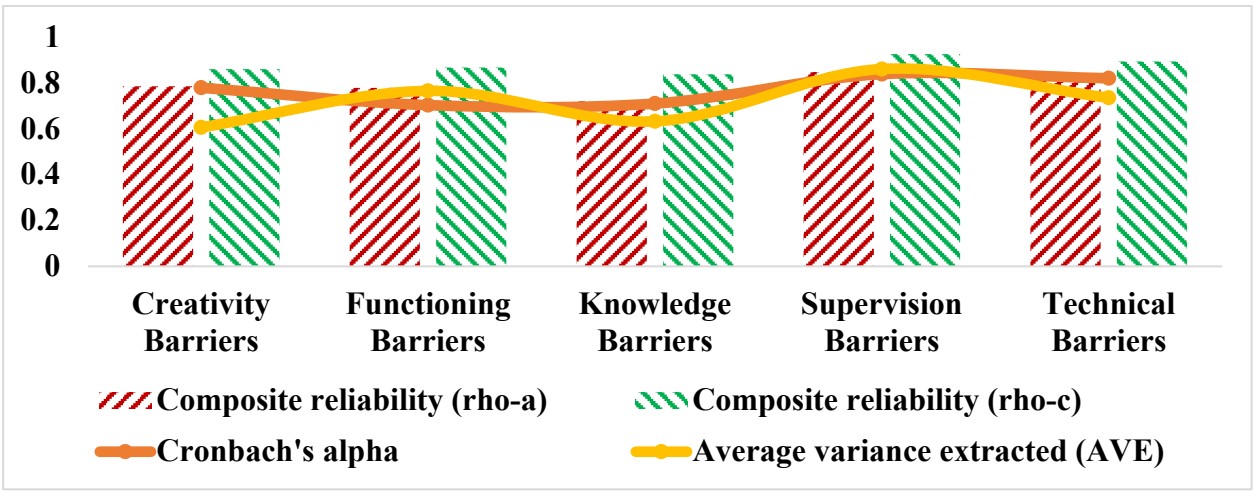

**Figure 4.** Trend of reliability and validity statistics.

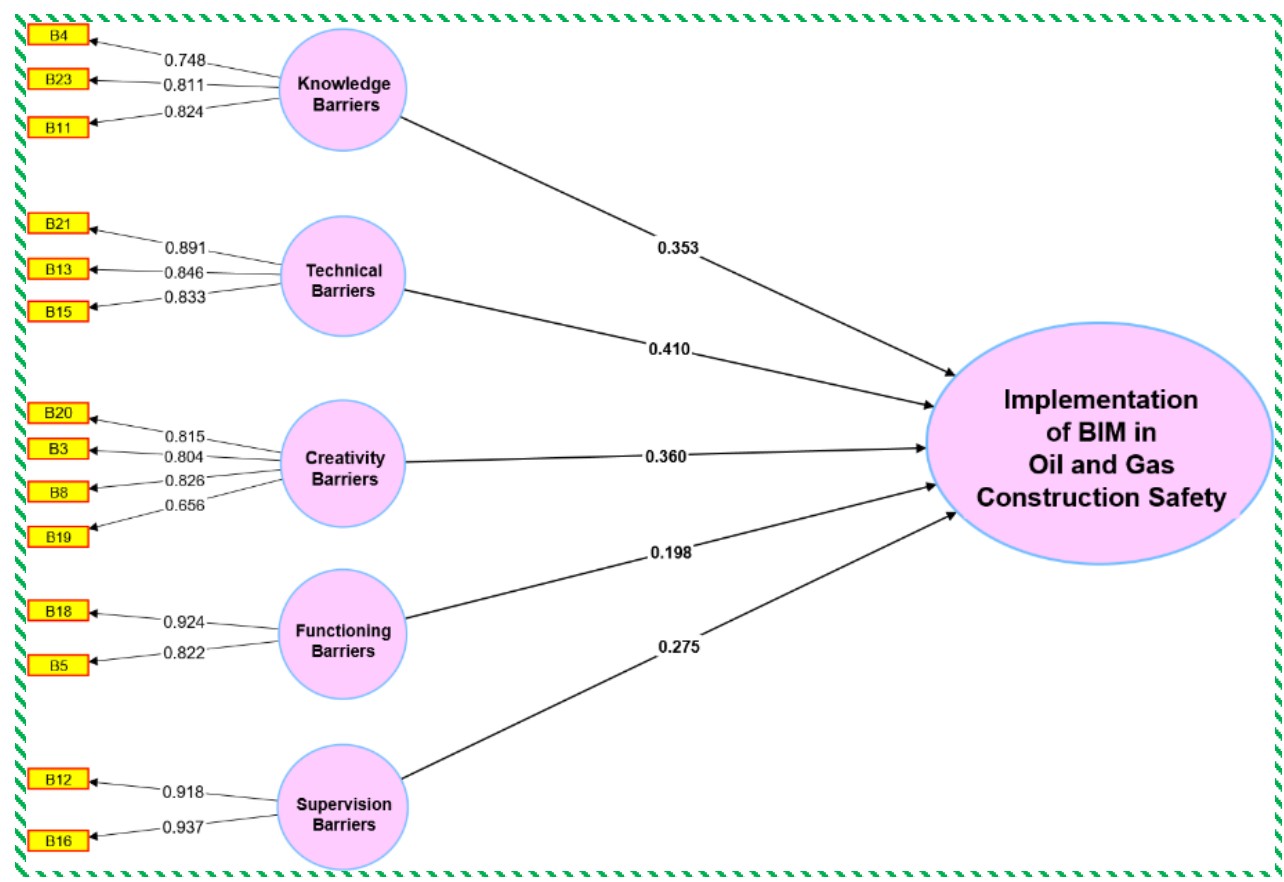

**Figure 5.** Model with path coefficients.

Second Order Analysis

The Heterotrait-Monotrait (HTMT) ratio analysis, which measures the discriminant validity of the constructs, yielded the data shown in Table 5. How much a construct has in common with other constructs rather than its indicators is quantified by the HTMT ratios. The HTMT ratio between two constructs should be less than 0.9 to be considered discriminantly valid. The data show that all the HTMT ratios are lower than 0.9, suggesting that the constructs in question may be considered separate [2,39]. This indicates that the discriminant validity of the five constructs is sufficient. The greatest HTMT ratio of 0.503 was found between technical barriers and knowledge barriers, indicating a significant

overlap between the two concepts. Yet, this is still below the cutoff of 0.9, as is needed to indicate that the two conceptions are different enough to be treated as independent variables in the analysis [16,82]. Overall, the findings of the HTMT analysis provide credence to the constructs' discriminant validity, suggesting that each of the five measures is a unique facet of the challenges to BIM deployment in the context of risk management for oil and gas construction projects in Malaysia.

**Table 5.** HTMT analysis results.

| Constructs | Creativity Barriers | Functioning Barriers | Knowledge Barriers | Supervision Barriers | Technical Barriers |
|---|---|---|---|---|---|
| Creativity Barriers | | | | | |
| Functioning Barriers | 0.241 | | | | |
| Knowledge Barriers | 0.263 | 0.187 | | | |
| Supervision Barriers | 0.204 | 0.132 | 0.283 | | |
| Technical Barriers | 0.471 | 0.179 | 0.503 | 0.218 | |

The discriminant validity of the constructs was evaluated using the Fornell–Larcker criteria, the results of which are shown in Table 6. The square root of each construct's average extracted variance (AVE) is shown on the diagonal. Values beyond the diagram's diagonal show correlations among the constructs. When the correlations between one concept and another are lower than the square root of the AVE for that construct, we have evidence of discriminant validity [21,24]. According to Table 6, all constructs are legitimate in terms of discriminant validity since the square root of the AVE for each construct is higher than its correlations with other constructs. Creativity barriers, for instance, have a higher AVE (0.778) and a stronger association (0.194) with other constructs than knowledge barriers (0.194), supervision barriers (0.165), or technical barriers (0.383). The findings point to the constructs' reliability and validity as measures of the latent variables they reflect. As shown by their discriminant validity, the constructs are good measures of separate elements of the challenges associated with implementing BIM in the context of risk management for oil and gas construction projects in Malaysia.

**Table 6.** Fornell and Larcker statistics.

| Constructs | Creativity Barriers | Functioning Barriers | Knowledge Barriers | Supervision Barriers | Technical Barriers |
|---|---|---|---|---|---|
| Creativity Barriers | 0.778 | | | | |
| Functioning Barriers | 0.194 | 0.875 | | | |
| Knowledge Barriers | 0.196 | 0.047 | 0.795 | | |
| Supervision Barriers | 0.165 | 0.11 | 0.229 | 0.927 | |
| Technical Barriers | 0.383 | 0.132 | 0.393 | 0.183 | 0.857 |

Table 7 displays the correlations between the five discovered components and the total number of BIM implementation difficulties (creativity barriers, functioning barriers, knowledge barriers, supervision barriers, and technical barriers). Good convergent validity may be inferred from the table since most items have strong loadings on their respective constructs [20,37]. Yet, there may be cross-loading concerns since some things have greater loadings on other structures. For instance, the knowledge barriers construct has a lower loading for B5 than the functional barriers construct. The table provides evidence that the selected items are legitimate measures of the constructs they are meant to assess; nevertheless, more research is required to address cross-loading difficulties.

**Table 7.** Cross-loadings of all BIM implementation challenges.

| | Creativity Barriers | Functioning Barriers | Knowledge Barriers | Supervision Barriers | Technical Barriers |
|---|---|---|---|---|---|
| B20 | **0.815** | 0.203 | 0.079 | 0.105 | 0.26 |
| B3 | **0.804** | 0.058 | 0.117 | 0.138 | 0.274 |
| B8 | **0.826** | 0.186 | 0.21 | 0.104 | 0.361 |
| B19 | **0.656** | 0.149 | 0.192 | 0.171 | 0.283 |
| B5 | 0.091 | **0.822** | −0.095 | 0.05 | 0.186 |
| B18 | 0.225 | **0.924** | 0.133 | 0.128 | 0.069 |
| B4 | 0.179 | 0.073 | **0.748** | 0.056 | 0.215 |
| B23 | 0.103 | −0.005 | **0.811** | 0.185 | 0.44 |
| B11 | 0.192 | 0.051 | **0.824** | 0.282 | 0.266 |
| B12 | 0.118 | 0.094 | 0.186 | **0.918** | 0.157 |
| B16 | 0.184 | 0.108 | 0.236 | **0.937** | 0.18 |
| B13 | 0.319 | 0.049 | 0.299 | 0.123 | **0.846** |
| B15 | 0.277 | 0.037 | 0.327 | 0.162 | **0.833** |
| B21 | 0.38 | 0.229 | 0.378 | 0.181 | **0.891** |

All of the variables that were eliminated in the study's EFA and SEM phases are summarized in Table 8. Table 1 reveals that during the EFA (pilot) stage, elements B2, B10, B17, B22, and B24 were removed, whereas elements B7, B14, B1, and B6 were removed during the SEM (main) stage. Deletions of factors during exploratory factor analysis (EFA) imply that those factors did not contribute enough to overall variance to be retained as full factors. The researchers excluded these from the final analysis to strengthen their confidence in the validity of the remaining constructs [44,50]. During the SEM phase, factors may be eliminated if it is determined that their absence would result in a better overall model fit. The researchers probably threw these out to boost the final model's reliability. Researchers removed these variables after seriously considering the constructs' reliability and validity and the overall model's robustness and accuracy.

**Table 8.** Summary of deleted factors.

| Variable | Status | Status |
|---|---|---|
| B2 | EFA (Pilot) | Deleted |
| B10 | EFA (Pilot) | Deleted |
| B17 | EFA (Pilot) | Deleted |
| B22 | EFA (Pilot) | Deleted |
| B24 | EFA (Pilot) | Deleted |
| B9 | EFA (Pilot) | Deleted |
| B7 | SEM (Main) | Deleted |
| B14 | SEM (Main) | Deleted |
| B1 | SEM (Main) | Deleted |
| B6 | SEM (Main) | Deleted |

*6.4. Path Analysis*

Based on their loading and VIF values, the factors are ranked in Table 9 to show how significantly they affect the group. Each construct, such as creativity, functionality, knowledge, supervision, and technology, has factors to consider. Lower VIF values indicate

less multicollinearity. Hence, they are used with the loading to determine the order. Group-wise, B13 (relative to technical challenges), B12 (relative to supervision hurdles), and B11 (relative to knowledge barriers) score best in terms of effect. This indicates that these factors are the most crucial in determining the difficulty of implementing BIM [21,48]. Functional challenges (B5 and B18) have the lowest group impact score, suggesting they have the smallest effect on BIM implementation difficulties. Overall, this ranking sheds light on which factors are more consequential in impacting BIM implementation difficulties, which may direct future study and practice toward more effective solutions.

**Table 9.** Group impact ranking.

| Variables with Constructs | Loading | VIF | Group Impact Ranking |
|---|---|---|---|
| B11 $\leq$ Knowledge Barriers | 0.824 | 1.439 | |
| B23 $\leq$ Knowledge Barriers | 0.811 | 1.388 | Rank 3 |
| B4 $\leq$ Knowledge Barriers | 0.748 | 1.345 | |
| B13 $\leq$ Technical Barriers | 0.846 | 1.83 | |
| B15 $\leq$ Technical Barriers | 0.833 | 1.725 | Rank 1 |
| B21 $\leq$ Technical Barriers | 0.891 | 2.01 | |
| B12 $\leq$ Supervision Barriers | 0.918 | 2.08 | |
| B16 $\leq$ Supervision Barriers | 0.937 | 2.08 | Rank 4 |
| B8 $\leq$ Creativity Barriers | 0.826 | 1.677 | |
| B3 $\leq$ Creativity Barriers | 0.804 | 2.121 | |
| B19 $\leq$ Creativity Barriers | 0.656 | 1.262 | Rank 2 |
| B20 $\leq$ Creativity Barriers | 0.815 | 2.139 | |
| B5 $\leq$ Functioning Barriers | 0.822 | 1.415 | |
| B18 $\leq$ Functioning Barriers | 0.924 | 1.415 | Rank 5 |

Path analysis was performed to verify the hypothesis about the connections between the five identified challenges to BIM implementation and the use of BIM in building projects. The findings are shown in Table 10. Standardized regression coefficients (β), their standard errors (SEs), *t*-values, *p*-values, and the variance inflation factor are all shown in the table for each possible direction of travel (VIF). The data show that all five challenges have a favorable correlation with BIM adoption. Each hurdle has a moderate to substantial impact on BIM adoption, as shown by the coefficients range (0.218–0.410). The impact of challenges to creativity on BIM adoption is the largest (β = 0.360, *p* = 0.001), followed by those to technical knowledge (β = 0.410), supervision (β = 0.277, *p* = 0.001), and functional efficiency (β = 0.198, *p* = 0.001). Multicollinearity is not a major problem in the model since all VIF values are less than 2.5. Overall, the findings imply that construction companies should prioritize removing the identified hurdles to BIM deployment to ensure the success of building projects [24,37]. Providing employees with training and support to boost their creativity and technical skills, fostering better communication and collaboration among project team members to reduce knowledge and functioning barriers, and supervising the implementation of BIM effectively are all examples of potential strategies for overcoming these challenges. Figure 6 presents the *p*-values for all the constructs and factors, whereas Figure 7 presents the t-stat for all links in the model.

Predictive relevance, or the model's capacity to accurately foretell new events given existing data, is shown in Table 11. The sum of squares explained (SSO), the sum of squares error (SSE), and Q2 are included in the table. Variability in the dependent variable (BIM implementation in construction) described by the model is denoted by SSO, whereas SSE denotes that which the model does not explain. The SSO of 3,933,000 indicates that the model can account for much of the dependent variable's variation. A model's predictive

accuracy, as measured by $Q^2$, is the fraction of variability in the dependent variable that can be explained by the model when tested using cross-validation. With a $Q^2$ of 0.245, the model has a fair amount of predictive accuracy. Values in Table 11 indicate that the model has a strong predictive ability and may be used to accurately anticipate the application of BIM in construction based on the identified challenges [21,82]. Predictive relevance is essential for gauging a model's success, but other metrics, such as goodness of fit and model complexity, should also be considered.

Every model construct and its relative weight and effectiveness are summarized in Table 12. The relevance score shows the importance of effectively adopting BIM in construction. In contrast, the performance score reflects how successfully the organization addresses the construct. The table shows that the organization is doing well in tackling the concept of functioning barriers, with a score of 62.776. Nonetheless, its relatively low relevance score of 0.23 suggests that it is less vital than other structures to achieving BIM's goals.

**Table 10.** Path analysis results.

| Path | β | SE | *t*-Values | *p*-Values | VIF |
|------|-----|-----|-----|-----|-----|
| Creativity Barriers ≥ Implementation of BIM in Construction | 0.360 | 0.019 | 20.725 | <0.001 | 1.213 |
| Functioning Barriers ≥ Implementation of BIM in Construction | 0.198 | 0.014 | 16.051 | <0.001 | 1.05 |
| Knowledge Barriers ≥ Implementation of BIM in Construction | 0.353 | 0.02 | 18.998 | <0.001 | 1.222 |
| Supervision Barriers ≥ Implementation of BIM in Construction | 0.275 | 0.015 | 14.929 | <0.001 | 1.083 |
| Technical Barriers ≥ Implementation of BIM in Construction | 0.410 | 0.024 | 15.341 | <0.001 | 1.347 |

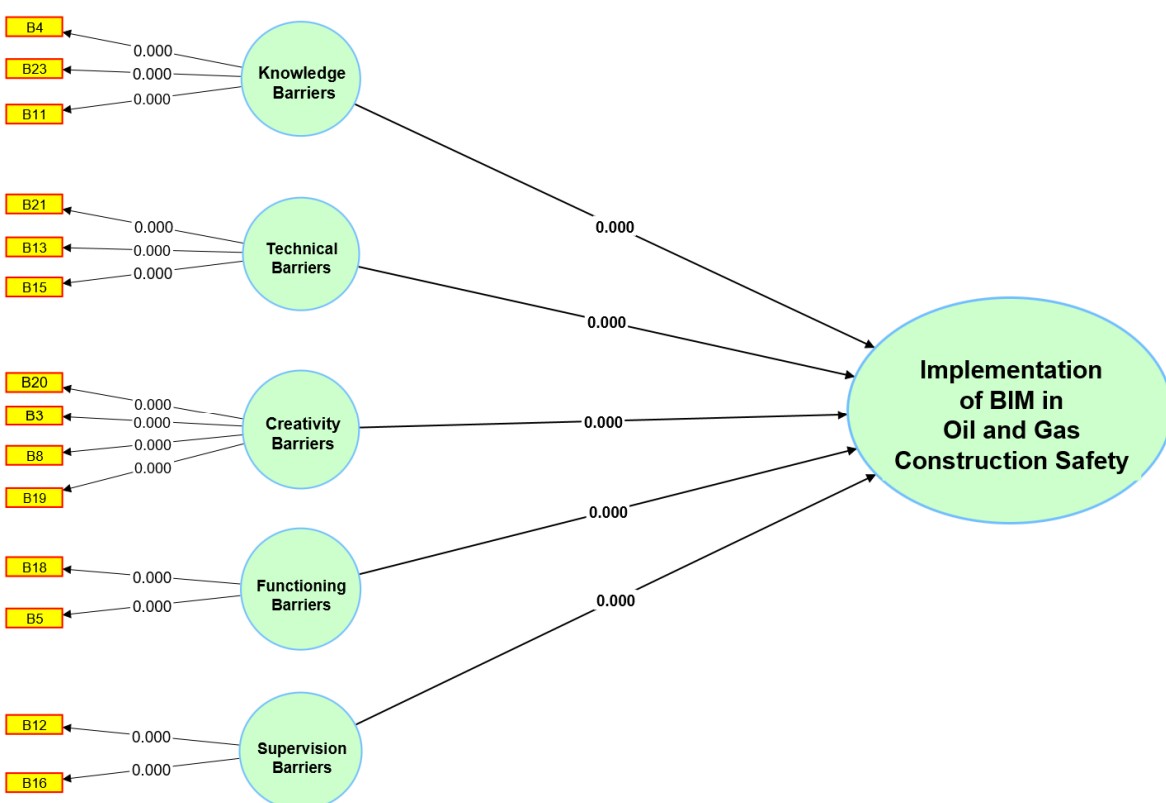

**Figure 6.** Model indicating the significance of challenges with constructs and constructs with latent variable.

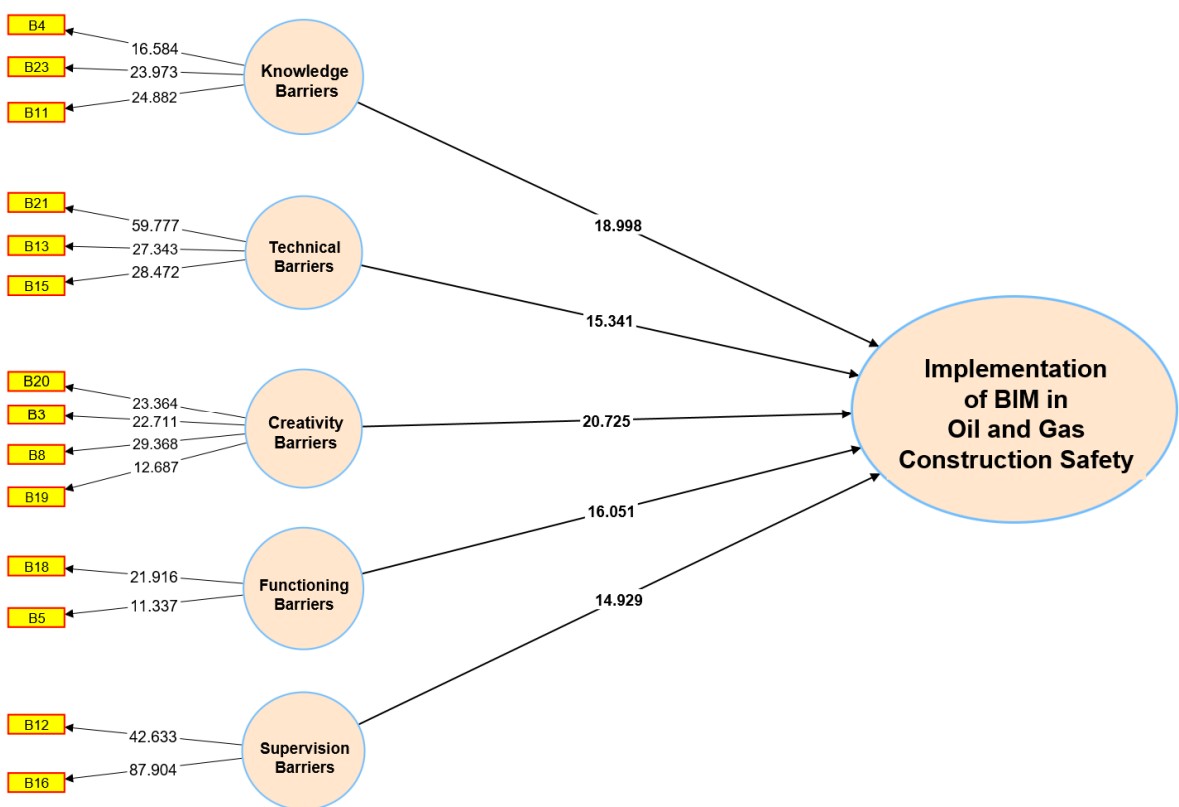

**Figure 7.** Model indicating t-stat of the challenges and constructs.

**Table 11.** Predictive relevance of the model.

| SSO | SSE | $Q^2$ |
|---|---|---|
| 3933.000 | 2968.574 | 0.245 |

**Table 12.** Importance and performance of constructs in the model.

| Construct | Performance | Importance |
|---|---|---|
| Creativity Barriers | 56.127 | 0.388 |
| Functioning Barriers | 62.776 | 0.23 |
| Knowledge Barriers | 53.938 | 0.388 |
| Supervision Barriers | 40.213 | 0.229 |
| Technical Barriers | 45.271 | 0.368 |

Nonetheless, with a performance score of just 40.213, the construct of supervision barriers requires more attention from the organizations. It has a high importance score of 0.229, indicating its significance in achieving BIM's goals. With moderate to high scores, the other constructs—creativity barriers, knowledge barriers, and technical barriers—show their significance to the BIM adoption process. Nonetheless, their performance ratings are mediocre, suggesting that more work is required to overcome these challenges [44,58]. This table may help the company solve the many challenges to BIM deployment by prioritizing the constructs with higher relevance ratings and lower performance scores.

*6.5. Model Validation*

Table 13 presents the results of an expert validation of a statistical model developed to evaluate the obstacles associated with the application of BIM for the risk management of oil

and gas construction projects in Malaysia. The average replies to the validation questions indicate that the recommended essential criteria may be employed, and the 19 responses validate the model's concept, objective, and conclusions. This research has considerable truth, and the structural models it generates are both conventional and generic.

**Table 13.** Model validation respondents.

| Respondent | 1 | 2 | 3 | 4 | 5 | 6 | 7 | 8 | 9 | 10 | 11 | 12 | 13 | 14 | 15 | 16 | 17 | 18 | 19 | Mean |
|---|---|---|---|---|---|---|---|---|---|---|---|---|---|---|---|---|---|---|---|---|
| Q1 | 4 | 3 | 4 | 5 | 4 | 3 | 5 | 3 | 5 | 4 | 5 | 5 | 4 | 4 | 5 | 4 | 4 | 4 | 5 | 4.21 |
| Q2 | 4 | 3 | 4 | 3 | 4 | 3 | 5 | 4 | 4 | 3 | 3 | 5 | 5 | 5 | 5 | 5 | 5 | 5 | 5 | 4.21 |
| Q3 | 4 | 5 | 4 | 5 | 3 | 3 | 4 | 5 | 5 | 4 | 5 | 4 | 5 | 4 | 5 | 4 | 4 | 4 | 4.32 |
| Q4 | 5 | 4 | 4 | 4 | 5 | 5 | 5 | 5 | 2 | 5 | 3 | 4 | 4 | 5 | 5 | 4 | 3 | 5 | 5 | 4.32 |
| Q5 | 5 | 4 | 5 | 4 | 4 | 4 | 5 | 5 | 5 | 5 | 5 | 5 | 5 | 4 | 4 | 4 | 4 | 3 | 5 | 4.47 |

Figure 8 depicts the final model created for assessing the obstacles associated with using BIM for the risk management of oil and gas construction projects in Malaysia. The concept is vital to the construction industry because it enables clients and contractors to execute oil and gas construction projects to a defined level of safety while protecting their respective benefits. Engineers, project managers, quantity surveyors, and companies may all benefit from the model's data. In addition, this technique guarantees that contractors strive to maintain their competitive advantage. Many respondents agreed with the optimistic conclusions of the survey.

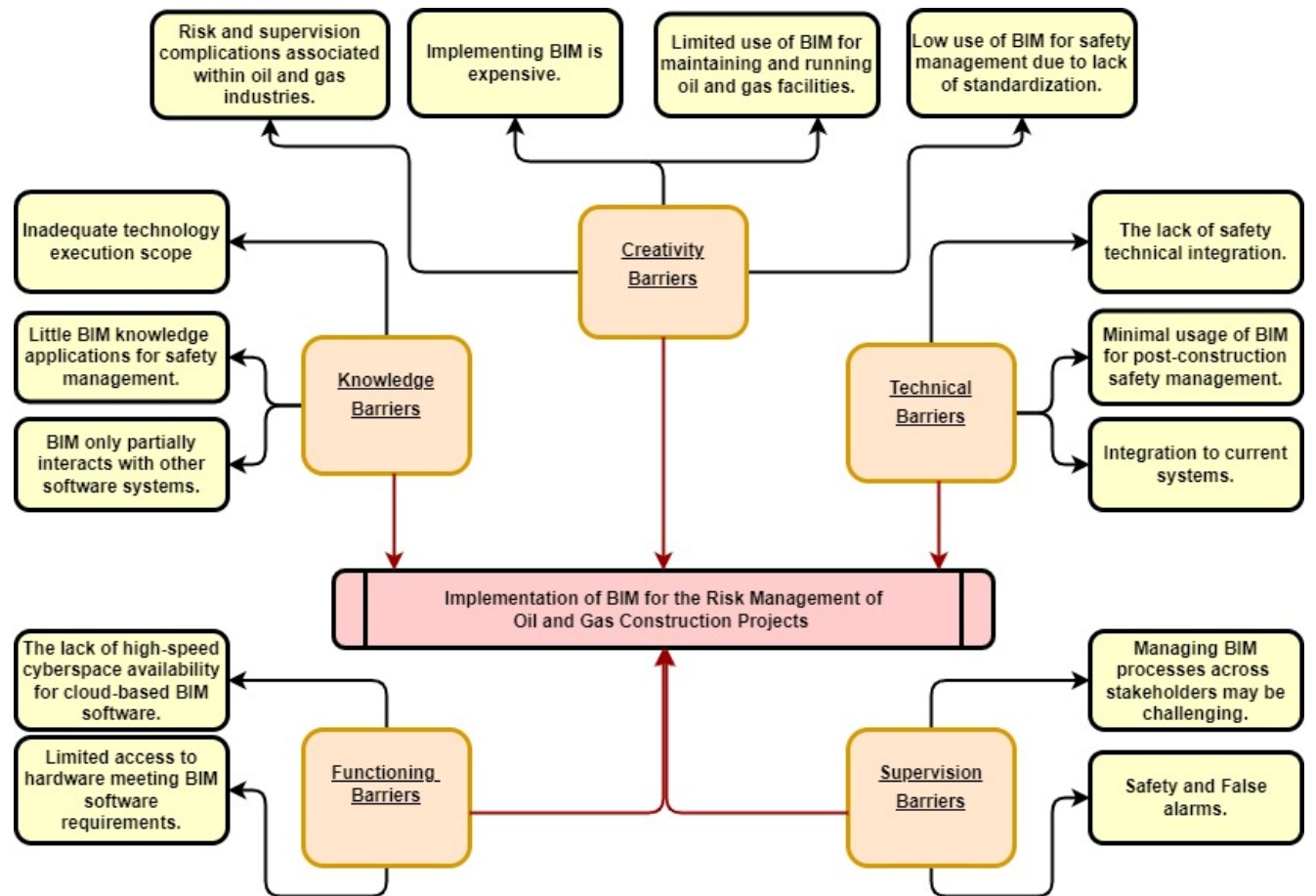

**Figure 8.** Final model generated by performing all the statistical analyses.

## 7. Discussion

The knowledge barriers formative construct ($\beta$ = 0.353, $t$ = 18.998, $p$ = 0.000) includes B23, "Inadequate technology execution scope," B11, "Little knowledge of BIM applications for safety management," and B4, "BIM software only partially interacts with other software systems." Path analysis shows that the knowledge barriers construct greatly influences BIM for risk management in Malaysian oil and gas development projects. Factors like limited familiarity with BIM applications for safety management and only a partial link between BIM software and other software systems are part of the build. Evidence for H1, that challenges in the knowledge barriers construct significantly influence BIM implementation for risk management in the context of oil and gas construction projects in Malaysia, can be seen in the high loading coefficient ($\beta$ = 0.353) and substantial $t$-value ($t$ = 18.998, $p$ = 0.000). As a result of these results, it is clear that action must be taken to remove the identified knowledge hurdles to BIM's increased use in the construction sector [20,24]. Therefore, this is something that businesses should strive for. Organizations may better manage risks in oil and gas construction projects by using BIM if they remove the knowledge challenges that prevent them from doing so. Based on the results, the hypothesis, "H1: The challenges in knowledge barriers construct strongly affects BIM implementation for risk management of oil and gas construction projects in Malaysia", is fully validated.

The technical barriers formative construct ($\beta$ = 0.410, $t$ = 15.341, $p$ = 0.000) includes B21, "The lack of safety technical integration", B13 "Minimal usage of BIM for post-construction safety management", B15 "Integration with current systems". The findings indicate that technological challenges greatly affect Building Information Modeling for risk management in Malaysian oil and gas development projects. There are concerns about integrating the technical aspects of safety, using BIM for post-construction risk monitoring, and incorporating existing systems into the design. In light of these results, it is clear that removing technological hurdles is essential for BIM adoption in the oil and gas sector in Malaysia, particularly in the context of risk management. Practitioners in the field may utilize this data to focus their efforts better to remove technical hurdles and increase BIM adoption on their projects. Based on the results, the hypothesis, "H2: Challenges in the technical barriers construct strongly affect BIM implementation for the risk management of oil and gas construction projects in Malaysia", is fully validated.

The creativity barriers formative construct ($\beta$ = 0.360, $t$ = 20.725, $p$ = 0.000) includes B20, "Risk and supervision complications associated with oil and gas industries," B3, "Low use of BIM for safety management due to lack of standardization," B8 "Implementing BIM is expensive," and B19 "Limited use of BIM for maintaining and running oil and gas facilities." The findings demonstrate that the formative construct of creative barriers significantly affects the use of BIM for risk management on oil and gas development projects in Malaysia. Risk and oversight difficulties, a lack of standardization, high implementation costs, and a lack of BIM application in facility maintenance and operations all figure into this framework. There may be challenges to original thinking and new approaches to problems needed to put BIM to good use in the building sector. The results of this study support the idea that removing these challenges would increase the industry's use of BIM, improving safety and risk management [2,82]. Based on the results, it is clear that more work has to be done to standardize BIM methods and lower the cost of implementation to increase its widespread use. Based on the results, the hypothesis, "H1: Challenges in the creativity barriers construct strongly affect BIM implementation for the risk management of oil and gas construction projects in Malaysia", is fully validated.

The functioning barriers formative construct ($\beta$ = 0.189, $t$ = 16.051, $p$ = 0.000) includes B5, "Limited access to hardware meeting BIM software requirements," and B18, "The lack of high-speed cyberspace availability for cloud-based BIM software." According to the findings of the route analysis, the functional barriers construct significantly improves the likelihood of using BIM for risk management in Malaysian oil and gas building projects. The build incorporates hardware and cyberspace availability factors essential to BIM software's operation. As shown by the coefficient of 0.189, there is an increase in BIM

adoption for risk management of 0.189 units for every one-unit rise in functional barriers. Regarding BIM's use in risk management, the functional barriers construct fares rather well, with a performance score of 62.776, as shown by the importance-performance analysis. Stakeholders see this as important, as the 0.23 significance score indicates.

For this reason, resolving issues with the working barriers construct is essential to using BIM for risk management on oil and gas building projects in Malaysia [20,21]. Many strategies might be implemented to solve these challenges, such as expanding access to hardware and the availability of high-speed cyberspace. Based on the results, the hypothesis "H1: Challenges in the functioning barriers construct strongly affect BIM implementation for the risk management of oil and gas construction projects in Malaysia", is fully validated.

The supervision barriers formative construct ($\beta = 0.275$, $t = 14.929$, $p = 0.000$) includes B12, "Managing BIM processes across various stakeholders may be challenging," and B16, "Safety and false alarms." The findings show that supervision-related hurdles significantly affect the use of BIM for risk management in Malaysian oil and gas building projects. According to the results, it may be difficult to effectively deploy BIM for risk management due to the difficulty of managing BIM procedures among different stakeholders. Major impediments to using BIM include concerns about safety and false alarms. Thus, efficient risk management needs to ensure that the data created by BIM is accurate and trustworthy [24,37]. The findings, taken as a whole, highlight the need to remove these supervisor-related challenges to BIM deployment and establish efficient risk management in Malaysia's oil and gas construction projects. Based on the results, the hypothesis, "H1: Challenges in the supervision barriers construct strongly affect BIM implementation for risk management of oil and gas construction projects in Malaysia", is fully validated.

### 7.1. Implications

The theoretical and practical implications of this study's results for using BIM for risk management in Malaysia's oil and gas construction sector are substantial. Theoretically, this research sheds light on the challenges that prevent the widespread use of BIM for risk management in the oil and gas construction sector. Results indicate that the five hurdles strongly predict using BIM for risk management: creative, functioning, knowledge, supervision, and technical. This shows how crucial it is to remove these challenges before Malaysia's oil and gas construction sector can successfully utilize BIM for risk management. In addition, this research contributes to the expanding literature on the use of BIM for risk management by highlighting the challenges that must be overcome. Results from this study inform efforts to use BIM for risk management in other sectors and nations of the construction industry. From a practical standpoint, this study's results may be utilized to direct policymakers and practitioners in Malaysia's oil and gas construction sector toward solutions to the challenges that prevent BIM from being effectively employed for risk management. Training and education initiatives for stakeholders to expand their understanding of BIM applications for safety management are one way to overcome the knowledge hurdles. However, the technological constraints may be overcome by integrating BIM software with existing systems and investing in and enhancing technical infrastructure. In addition, professionals in the field may utilize the study's findings to plan how to remove the challenges to BIM's usage in risk management. Many methods have been proposed to overcome the limitations of creativity and supervision. Finally, this research sheds light on the challenges that must be overcome before Malaysia's oil and gas construction sector can successfully use BIM for risk management. This research has both theoretical and practical consequences. It may help policymakers and practitioners devise solutions to overcome these challenges and assure the effective use of BIM for risk management.

### 7.2. Managerial Recommendations

The results of this research provide the following managerial suggestions for advancing the use of BIM in Malaysian oil and gas construction projects for risk management.

Improving construction workers' knowledge and abilities may be as simple as giving them a crash course on BIM and how it can be used to manage risk better. This may be useful in removing challenges associated with a need for more understanding or resources for BIM. Promoting standardization and compatibility of BIM software with other systems used in the construction sector might enhance BIM deployment. This may assist in getting over the conceptual and technological hurdles that have been holding back BIM. With proper stakeholder coordination and open lines of communication, BIM deployment may proceed with little interference from supervisors. Effective adoption of BIM requires careful management of BIM procedures across several stakeholders. Increasing connectivity to high-speed networks and making BIM-compatible hardware more widely available are two examples of how cyberinfrastructure development may remove challenges to efficiency. BIM may have financial and operational advantages; a cost-benefit analysis can help you weigh the two sides. This may be useful in getting beyond the creative blocks that have been holding back BIM thus far. Constant refinement is important for BIM implementation, by constantly monitoring and evaluating our progress. Maintaining an efficient implementation process is crucial for efficient risk management. With these suggestions, Malaysia's oil and gas construction projects will be safer, more efficient, and of higher quality.

### 7.3. Limitations

The research had significant methodological limitations that may have affected the findings. The researchers' reliance on exploratory factor analysis (EFA) and structural equation modeling (SEM) impacted the study's findings and breadth. Confirmatory factor analysis (CFA) was not used in this research, but it may be used to corroborate these results and offer a fuller picture. Another caveat is that the research only looked at one sector of the Malaysian economy—the oil and gas sector. The limitations of SEM in the context of the topic include the potential for model misspecification due to the complexity of the oil and gas construction industry, difficulty in accurately measuring and defining variables related to risk management, and the need for a large and diverse sample size to ensure the generalizability of findings. Additionally, the interpretation of results from SEM requires a certain level of expertise in statistical analysis. This limits the findings' applicability to other contexts, such as different sectors or geographical areas. To give a more in-depth examination of the variables influencing the use of BIM for risk management, future studies may broaden the research scope to include additional sectors or areas. In addition, this research relied on data collected through self-administered questionnaires, which may have introduced some degree of response bias into the overall findings. Interviews and focus groups might be useful in future research to learn more about the elements that influence the use of BIM for risk management. BIM's potential to enhance risk management in building projects deserves more investigation, particularly across sectors and geographies. In addition, research might evaluate the efficacy of various approaches to removing the challenges of using BIM.

## 8. Conclusions

In conclusion, this research looked at the difficulties associated with using BIM for risk management on oil and gas building projects in Malaysia. The research showed that five fundamental constructs greatly impacted the use of BIM for risk management: creative barriers, functional barriers, knowledge barriers, supervision barriers, and technical barriers. All of the predicted findings for each construct were confirmed. Findings imply that difficulties associated with these elements may impede BIM's use for risk management in the oil and gas construction sector. The research offers theoretical and practical suggestions on best using BIM to mitigate risks in Malaysia's oil and gas construction sector. It stresses the significance of resolving issues associated with these elements to ease the use of BIM for risk management. The methodological approach used in this study is one such constraint; future research may want to look at different options for verifying the results. Other elements that may affect the use of BIM for risk management on oil and gas construction

projects in Malaysia might be the subject of future study. In sum, this research adds to the body of knowledge by shedding light on the difficulties inherent in using BIM for risk management in Malaysia's oil and gas construction sector. The results may be used to make better decisions and assist in creating more efficient strategies to overcome the challenges blocking the industry's path to the widespread adoption of BIM.

**Author Contributions:** Conceptualization, A.W.; Methodology, A.W.; Software, I.O.; Validation, A.W. and R.A.G.-L.; Formal analysis, A.W.; Investigation, I.O.; Resources, I.O.; Data curation, I.O.; Writing—original draft, A.W.; Writing—review and editing, R.A.G.-L.; Visualization, A.W.; Supervision, R.A.G.-L.; Project administration, I.O.; Funding acquisition, R.A.G.-L. All authors have read and agreed to the published version of the manuscript.

**Funding:** This research received no external funding.

**Data Availability Statement:** Not applicable.

**Conflicts of Interest:** The authors declare no conflict of interest.

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
