# Peer review of "Challenges to the Implementation of BIM for the Risk Management of Oil and Gas Construction Projects: Structural Equation Modeling Approach"

_sustainability, doi:10.3390/su15108019_

Round 1
Reviewer 1 Report
This paper introduces Challenges to implementing BIM for the Risk Management of Oil and Gas Construction Projects in Malaysia: Structural Equation Modeling Approach. However, the significant observations of the reviewer are listed below, which must be addressed before publication.
0. Tile is too long. It must be within 15 words without abbreviations.
1. Highlights of this work must be included.
2. Long and vague Abstract. It must be truncated to be concise and self-informative.
3. The English of the paper is very poor, with many grammatical flaws, typos, long and vague sentences, and word repetitions.
4. Introduction is much more lecturing. Instead of giving the contribution, previous problems and also the lack of methods. What is the problem? Why is the previous model incomplete? What was their problem? Which part of previous work in the scientific borders has progressed? And why?
5. The last paragraph of the Introduction should be assigned to a summary of the applied method and bolded findings.
6. I cannot see any novelty in the applied model, as it follows the routine procedure claimed for Risk Management of Oil and Gas.
8. Simulation results and discussion section present results, no comparison with previous methods.
9. No comparative study with similar kinds of reports is already available. The authors made similar comparative studies with his/her simulation, which may not be considered worthwhile.
10. The results show that all models provide similar types of Risk Management of Oil and Gas.
Reviewer 2 Report
Please see comments in post-its in the attached pdf. I stopped reviewing at p.8 because the paper far exceeds the recommended wordcount. Please obtain advice from the editors.

Reviewer 3 Report
The manuscript entitled “Challenges to the Implementation of BIM for the Risk Management of Oil and Gas Construction Projects in Malaysia: Structural Equation Modeling Approach” proposes an analysis of the difficulties in using BIM in oil and gas construction projects in Malaysia. The article is interesting, the methodology is dense and the results consistent, however, these are some topics that I believe could be improved.
MAJOR COMMENTS:
1) ABSTRACT: The Abstract must be a faithful summary of the paper. It should address all the main aspects of the study, that is, contextualization, objective, methodology, and main results. The last three items are satisfactorily presented, but the contextualization (Lines 1 to 8) needs to be improved. In my opinion, the text is confusing and does not adequately represent the challenges of using BIM. Also, watch out for excessive repetition of sentences. The phrase "in Malaysia's oil and gas construction sector", despite being part of the study's title, was repeated 5 times only in the abstract, making it difficult to read.
2) Authors must promote an English check. Some sentences are repetitive and confusing and need rewriting. Here are a few examples:
- Abstract: “Because of the inherent dangers inherent in the oil and gas business, ...”
- Introduction (3rd paragraph): “It is essential to recognize that the frequency and severity of accidents and incidents in oil and gas construction projects accidents and incidents may vary ...”
- Introduction (3rd paragraph): “and mitigate possible hazards and safeguard the safety of employees ...”
- Methodology (1st paragraph): “given structural equation modeling considerable consideration”.
- Section 5.2 (1st paragraph): “... of observed variable variation explained ...”
- Section 5.2 (1st paragraph): “A greater number suggests that the variables adequately describe the variable.”
3) INTRODUCTION: The introduction section is a bit long, and the author ends up repeating the same information. For example:
- 8th paragraph: “This paper attempts to solve this research gap using a structural equation modeling technique to examine the barriers to BIM adoption for risk management in Malaysian oil and gas construction projects”.
- 9th paragraph: “This publication seeks to uncover the elements that impede the application of BIM and its interrelationships using structural equation modeling”.
- 11th paragraph: “This publication comprehensively explains the interrelationships between the many constraints that impede BIM adoption in the Malaysian oil and gas sector via structural equation modeling”.
In my opinion, the Introduction section should be a concise text that provides a fast initial understanding of the research. For this purpose, it must only consist of a brief introduction to the topic, delimit the research gap, the objective, highlight the novelty of the research, and provide a brief explanation of the methodology to be used. This way the author avoids adding unnecessary information. Therefore, I suggest that the Intro section be revised.
4) SECTION 2: In my opinion, this section needs improvement. The three paragraphs basically present statistics referring to three institutions (DOSH, IOGP, and BLS), not constituting a literature review. Therefore, I suggest that a more in-depth bibliographical review be added on the subject.
5) SECTION 3: The author mentions some studies that have already evaluated the use of BIM to improve safety management, but the information provided is very superficial, attesting that the studies have proven that BIM is interesting for this application. Therefore, I suggest that the author delve a little deeper into the results of these studies since they are a relevant reference for this research.
6) TEXT STRUCTURE: As Section 4 (Identification of challenges) is a result of the author's work, I suggest an inversion in the paper structure, so that the Methodology Section is presented before Section 4.
MINOR COMMENTS:
7) On your next submission, please number the lines to facilitate the review process.
8) I suggest including "Structural Equation Modeling" as a keyword, in order to increase the paper's visibility in the indexed databases if it is published.
9) In the first sentence of the Introduction Section: “The oil and gas sector is among the most difficult and dangerous in the world”.
Considering this is the opening sentence of the article, it is extremely generic and dispensable in my opinion. What would be a “difficult” sector? How dangerous is inserted in this context? Regarding financial risks? Safety? Market variations? All of them? I suggest that the opening sentence be more assertive.
10) INTRODUCTION SECTION. In the sentence: “The United States Occupational Safety and Health Administration (OSHA) reports that the oil and gas extraction business has a higher death rate than the average across all industries. In 2019, the mortality rate for oil and gas extraction was 9.2 per 100,000 full-time equivalent employees, compared to the industry average of 3.5 per 100,000 [1], [5]”.
Is this information related to construction and operation activities together? Or just operation activities? Care should be taken when comparing statistics from this sector with other industry sectors.
11) INTRODUCTION SECTION. In the sentence: “Construction projects using oil and gas contain combustible materials and gases that may ignite and cause fires and explosions [6]. In 2019, 12 percent of all deaths in the oil and gas business in the United States were caused by fires and explosions. Construction projects involving oil and gas necessitate operating at height, which may lead to falls and slides. In 2019, 16% of deaths in the oil and gas business in the United States were caused by falls and slips”.
Will this paper only cover the construction stage or also the operation stage? If the first option, the statistics covering the operation stage are not relevant to the study.
12) SECTION 4: The author must clearly state that the challenges described in Table 1 were obtained through a literature review.
13) All tables and figures must be cited in the text as close as possible to where they are located. Please check all tables and figures.
14) Although it is not within the scope of the review task, I would like to suggest that authors observe the journal's “Instructions for Authors” in order to prepare the manuscript according to its guidelines. References cited throughout the text do not show the year of publication or their numbering. For example:
- Section 3 (1st paragraph): “Elwany & El-sharkawy, and Mohd Hanafiah et al., for...”
- Section 5 (1st paragraph): “A critical review reveals that Ajmal, Bin Isha, et al., and Leth et al., have...”
- Section 5.1 (Last paragraph): “According to Mohd Hanafiah et al., the ...”
- Section 5.2 (Last paragraph): “This method assisted Sohrabi & Noorzai in...”
Please check all the text.
15) SECTION 5.3.1: “For all study forms, values more than 0.70 and greater than 0.60 were deemed adequate”. I suggest rewriting the sentence, because a value greater than 0.6 is intrinsically greater than 0.7.
16) All equations must be numbered and properly cited in the text next to where they are located.
17) Please, increase the font size in Figure 3.
Reviewer 4 Report
What is meant by BIM based risk management is not explained. There are several options how BIM can be used to manage risk and these need to be explained in an additional paragraph in the work How do these work? What are the options? What is the current state of the art? This should be appropriately referenced. A BIM model alone will not act as a hazard warning system. How are risks integrated with assets and processes? What hazard awareness methodologies can be used? Otherwise the paper has high academic rigour. Figure 8 is particularly informative
Round 2
Reviewer 1 Report
My concerns are addressed. No further comments.
Author Response
The authors are very thankful to the reviewer for considering the manuscript, and all provided suggestions helped in improving manuscript.
Reviewer 2 Report
My suggestions and recommendations have been addressed satisfactorily
Author Response
The authors are very thankful to the reviewer. All comments were helpful in addressing the deficiencies in manuscript.
Reviewer 3 Report
Congratulations to the authors for the changes implemented during the first review round. However, I consider that the article still needs a little attention to be published in this journal. Please check the following points, which were indicated in the first round of review and were not properly addressed.
1) COMMENTS FROM THE 1st ROUND OF REVIEW – “SECTION 2: In my opinion, this section needs improvement. The three paragraphs basically present statistics referring to three institutions (DOSH, IOGP, and BLS), not constituting a literature review. Therefore, I suggest that a more in-depth bibliographical review be added on the subject.”
COMMENTS FROM THE 2nd ROUND OF REVIEW: With the exception of Section 3, which talks about “BIM for Risk Management”, Section 2 is the only one that presents an attempt at a literature review. Considering that “Current Risk and Safety Management Concerns” constitutes a central point for the work, presenting a section with only some statistics in place of a literature review, in my opinion, is not acceptable for this journal.
Authors need to understand that there is a difference between a “literature review” and a “bibliometric literature review”. My comment concerns a “literature review”, which is essential for any research project, as it allows authors to reach the state of the art on the subject, presenting the main results of research already published on the topic, thus guaranteeing the novelty of the research. So, in my opinion this section should be improved. Furthermore, MDPI journals do not limit article length. Therefore, the justification of the authors regarding the impossibility of expanding the section is not valid.
2) COMMENTS FROM THE 1st ROUND OF REVIEW– “SECTION 3: The author mentions some studies that have already evaluated the use of BIM to improve safety management, but the information provided is very superficial, attesting that the studies have proven that BIM is interesting for this application. Therefore, I suggest that the author delve a little deeper into the results of these studies since they are a relevant reference for this research.”
COMMENTS FROM THE 2nd ROUND OF REVIEW: The authors' response left me confused. How did the indicated studies have not focused on identifying challenges of BIM implementation in oil and gas construction project risk management?
· “Numerous prior studies have investigated the potential advantages of BIM for enhancing safety management and risk reduction in the construction industry, particularly the oil and gas industry [32], [33].”
· “In addition, Annamalai et al. and Jagoda & Wojcik investigated the potential for BIM to improve the safety of oil and gas construction projects in the United Arab Emirates [36], [37].”
· “These studies illustrate the capability of BIM to improve safety management and risk reduction in the construction industry, particularly the oil and gas industry [38], [39].”
These are a few sentences taken from the text. This is a growing field of study in the technical literature, so if the authors believe that these papers are not enough to present a more complete literature review, I suggest that they search a little more in the indexed databases.
Reviewer 4 Report
The weaknesses in the structural modelling method could have been further discussed. Further explanation about why the literature sources where selected could have been included.
